# Intensifying rice production to reduce imports and land conversion in Africa

Shen Yuan [1], Kazuki Saito [2,3], Pepijn A. J. van Oort[4], Martin K. van Ittersum [5,6], Shaobing Peng [1] ✉ & Patricio Grassini [7] ✉

Africa produces around 60% of the rice the continent consumes, relying heavily on rice imports to fulfill the rest of the domestic demand. Over the past 10 years, the rice-agricultural area increased nearly 40%, while average yield remained stagnant. Here we used a process-based crop simulation modelling approach combined with local weather, soil, and management datasets to evaluate the potential to increase rice production on existing cropland area in Africa and assess cropland expansion and rice imports by year 2050 for different scenarios of yield intensification. We find that Africa can avoid further increases in rice imports, and even reduce them, through a combination of cropland expansion following the historical trend together with closure of the current exploitable yield gap by half or more. Without substantial increase in rice yields, meeting future rice demand will require larger rice imports and/or land conversion than now.

Rice cultivation in Africa has a long history dating back more than 3000 years in lowlands located in river deltas and inland valleys as well as uplands[1]. While African diets have historically relied on other staple crops, such as maize, sorghum, and cassava, there has been a progressive shift towards greater rice consumption due to economic growth and associated consumer preference for eating rice[2–4]. This shift in diets, coupled with a sharp population increase (+113%), has driven a substantial increase in rice consumption in Africa over the past 30 years[3,5] (Fig. 1). Higher demand for rice has been met through a parallel increase in rice domestic production and imports. However, rice area expansion, rather than yield increase, has been the primary driver for increasing domestic production, with cropland expanding by ca. 0.4 M ha per year over the past three decades[6]. At the same time, rice imports have increased steadily over the same period and currently account for around 40% of total rice consumption in Africa, representing one-third of all rice traded on the global market[4]. The average annual amount of imported rice between 2018 and 2020 was 25 million metric tons (Mt), which is worth US$7 billion given current global rice price[5,7].

Further increase in population and rice consumption per capita will more than double the African demand for rice by 2050, relative to year 2020, reaching around 150 Mt[3,8]. Without substantial yield increase, higher demand for rice will further increase reliance on costly imports and/or massive cropland expansion[6,9–11]. While reaching rice self-sufficiency is not necessarily a goal by itself, we note that achieving a reasonable level of self-sufficiency for main staple crops is desirable for countries with small monetary reserves to afford expensive food imports, as it is the case for most countries in Africa[10,12]. Furthermore, heavy reliance on grain imports makes countries vulnerable to price volatility and price shocks on the global market due to factors including trade taxes and restrictions, and political turmoil[2,13–16]. On the other hand, a low level of self-sufficiency could trigger massive expansion of rice area in Africa, which, in turn, leads to encroachment of natural ecosystems, and increased global warming potential due to

[1]National Key Laboratory of Crop Genetic Improvement, Hubei Hongshan Laboratory, MARA Key Laboratory of Crop Ecophysiology and Farming System in the Middle Reaches of the Yangtze River, College of Plant Science and Technology, Huazhong Agricultural University, Wuhan, Hubei 430070, China. [2]Africa Rice Center (AfricaRice), 01 B.P. 2551 Bouaké 01, Côte d'Ivoire. [3]International Rice Research Institute (IRRI), DAPO Box 7777 Metro Manila 1301, Philippines. [4]Wagfnoveningen Plant Research, Agrosystems Research, P.O. Box 16, 6700 AA Wageningen, the Netherlands. [5]Plant Production Systems Group, Wageningen University & Research, PO Box 430, NL-6700 AK Wageningen, the Netherlands. [6]Department of Crop Production Ecology, Swedish University of Agricultural Sciences, 75007 Uppsala, Sweden. [7]Department of Agronomy and Horticulture, University of Nebraska-Lincoln, Lincoln, NE 68583-0915, USA. ✉e-mail: speng@mail.hzau.edu.cn; pgrassini2@unl.edu

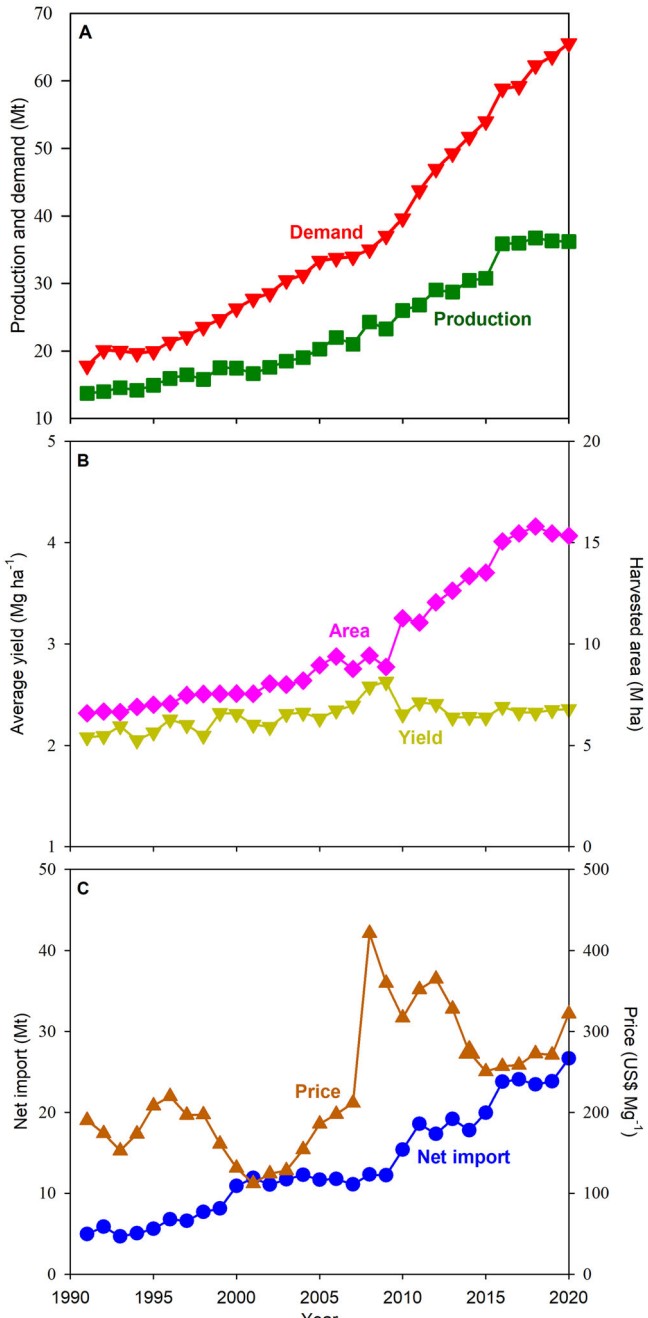

**Fig. 1 | Historical patterns in rice production, demand, imports, average yield, and price in Africa.** Trends in (**A**) rice demand and production, (**B**) average yield and harvested area, and (**C**) net import and rice price in Africa during the past 30 years (1991–2020). Rice demand was estimated based on annual regional rice production, import, export, and stock variation. Data are reported on a paddy-rice basis and are from USDA[5], FAO[6], and World Bank[7]. Source data are provided as a Source Data file.

land conversion and greenhouse gas emissions from new cropland[9,17]. Thus, increasing rice imports and/or expanding cropland area is not a sustainable pathway from an economic, socio-political, and environmental perspectives to meet domestic African demand for rice.

Crop intensification, that is, increasing productivity on existing cropland, can help reduce the need for costly food imports and/or relieve pressure on land conversion for agriculture[10,11,18–23]. In the case of rice in Africa, there is a clear opportunity for increasing production via intensification, given that average yield has remained largely

stagnant over decades and lower than that in other rice producing countries[10,11,19] (Fig. 1). For example, average rice yield in Africa is 2.4 Mg ha⁻¹, which is 33% lower than that achieved by farmers in Southeast Asia (4.3 Mg ha⁻¹)[6]. Such difference may be associated with different agronomic management in African rice systems, including insufficient nutrient supply, difficulties in controlling pests, weeds, and diseases, and less efficient soil and water management[19,24–26]. Likewise, differences in yields between Africa and other rice-producing regions may be associated with differences in the environmental condition and water regime. For example, while a large fraction of rice cropping systems in Asia and South America are irrigated, most rice in Africa is grown in rainfed lowland and upland environments[4,24,27]. In these environments, crops rely solely on rainfall amount and distribution during the growing season and (in lowland rice) capillary rise to meet their water requirements, which could lead to episodes of water deficit and/or excess. Hence, a pertinent question is how much room exists to increase current yields in Africa and to what degree yield intensification can help Africa achieve a reasonable level of rice self-sufficiency while reducing pressure on cropland expansion and reliance on food imports.

Reversing the trends and threats of cropland expansion and high import dependency requires science-based information and active promotion and investment in intensification of current rice cropping systems by national and regional policymakers in Africa[11]. Here we estimate the yield gaps, that is, the difference between yield potential and average farmers' yield, to assess the existing room for improving rice production through crop intensification, relying on crop modeling and extensive on-the-ground data collection, including weather, soil, and management data across 15 countries that collectively account for 10 million ha planted with rice and 80% of total rice production in Africa[6]. Subsequently, we assess future reliance on rice imports and cropland expansion for different scenarios of crop intensification. We discuss implications of our findings for prioritizing agricultural research and development (R&D) programs at regional, national, and subnational levels.

## Results

### Large spatial variation in yield potential and yield gap

Rice in Africa is grown across a wide range of agroecological zones (from temperate to humid tropics), environments (lowland, upland), and water regimes (rainfed, irrigated). Yield potential is determined by solar radiation, temperature, and length of the crop season in the case of irrigated crops, and by precipitation and soil type in the case of rainfed crops in lowland and upland environments. Here we focused on 20 country-water regimes accounting for 65% and 80% of rice harvested area and production in Africa, respectively (Supplementary Table 1). We estimated the yield potential for irrigated rice, rainfed lowland rice, and rainfed upland rice separately using a process-based crop simulation model based on long-term weather, dominant soil types, and local crop calendars, and accounting for differences in water supply in each system[28] (Supplementary Figs. 1 and 2; Supplementary Tables 2 and 3, see Methods). Our simulations accounted for the influence of weather and soil properties on yield potential and assumed no nutrient limitations and no yield reduction due to biotic factors.

Across all sites, water regimes, and environments, the area-weighted yield potential was 8 Mg ha⁻¹, varying widely across regions, from ca. 4 Mg ha⁻¹ in rainfed upland rice in West Africa up to 11 Mg ha⁻¹ in irrigated rice in the Nile delta in Egypt (Fig. 2). Average area-weighted yield potential of irrigated rice was higher than for rainfed rice (9.9 Mg ha⁻¹ versus 7.0 Mg ha⁻¹) and more stable across years as determined based on the coefficient of variation and downside risk analysis (Supplementary Table 3). There were also marked differences between rainfed upland and lowland rice. For example, average water-limited yield potential was higher (8.1 versus 5.6 Mg ha⁻¹) and more

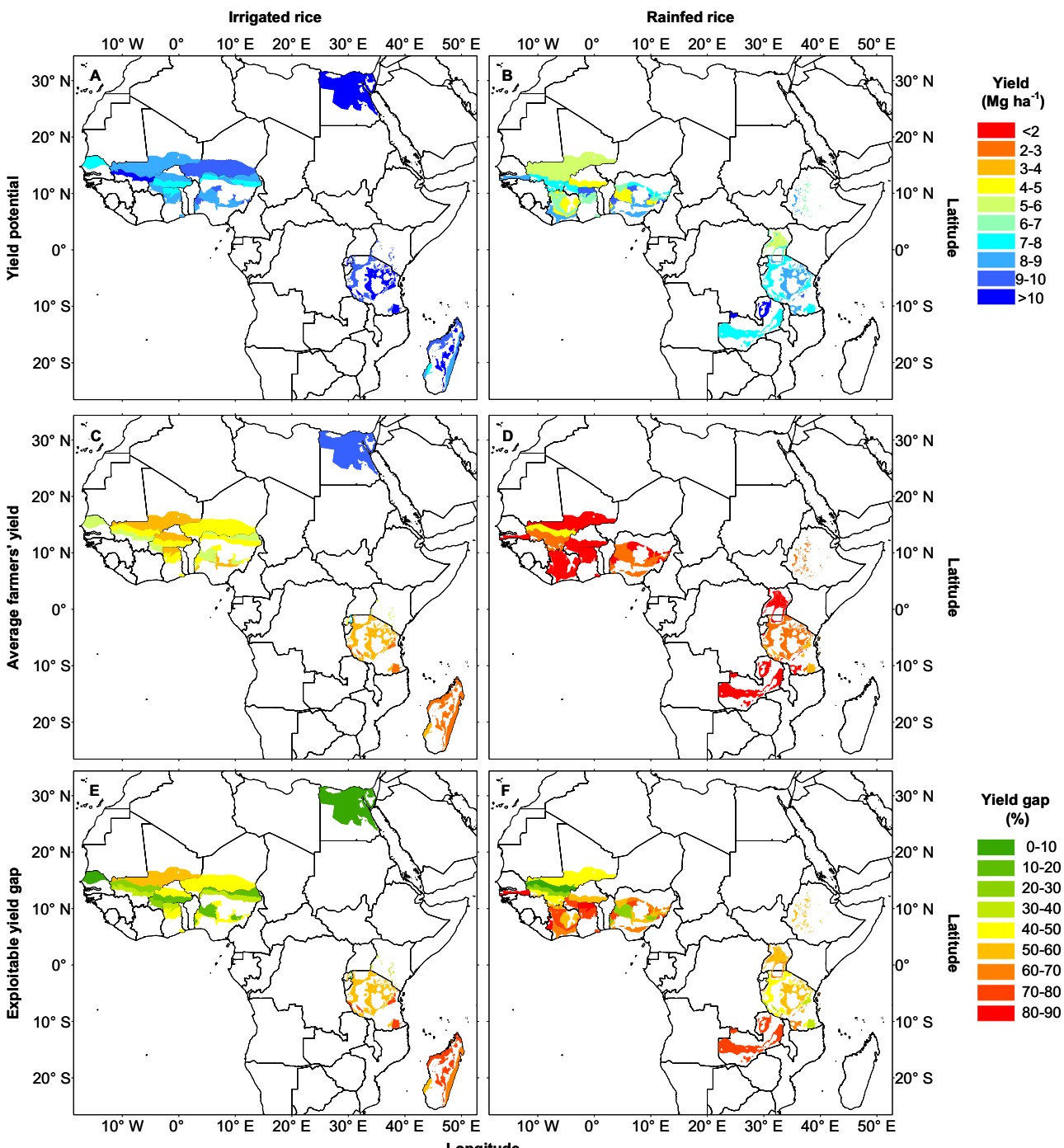

**Fig. 2 | Yield potential, average farmers' yield, and exploitable yield gap for irrigated and rainfed rice in Africa at the climate zone level.** Panels show (**A, B**) yield potential, (**C, D**) average farmers' yield, and (**E, F**) exploitable yield gap (as a percentage of attainable yield) for (**A, C, E**) irrigated and (**B, D, F**) rainfed rice across 15 rice-producing countries in Africa. See Methods for yield potential simulation and yield gap estimation. Source data are provided as a Source Data file. The base map was applied without endorsement using data from the Database of Global Administrative Areas (https://gadm.org/).

stable in rainfed lowland rice compared with rainfed upland rice (Supplementary Fig. 3). Average farmers' yield followed the same trends as yield potential but was considerably lower (Fig. 2). Across all sites, water regimes, and environments, the area-weighted actual yield was 2.9 Mg ha⁻¹, which represents only 36% of the average yield potential. Cross-validation with measured yields in well-managed crops in field trials in Africa, as well as with the yield potential estimated for analogous climate zones in the rest of the world, showed that our estimation of yield potential for Africa is robust (Supplementary Table 4).

Reaching yield potential is difficult as it requires copious amounts of inputs and labor and fine-tuning of soil and crop management practices. Thus, reaching 70-80% of yield potential is a more realistic target for farmers with reasonable access to inputs, markets, and technical information[29,30]. We estimated here the attainable yield as 80% and 70% of the simulated yield potential for irrigated and rainfed rice, respectively, and calculated the exploitable yield gap as the difference between attainable yield and actual farmers' yield. Across all sites, water regimes, and environments, the area-weighted exploitable yield gap represented 52% of the attainable yield (Fig. 2). All sites

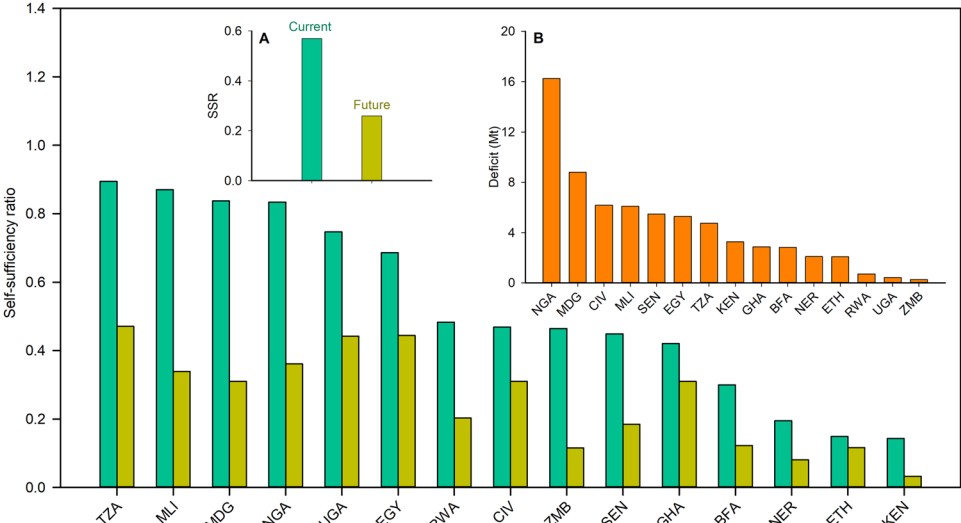

**Fig. 3 | The current (2018-2020) and future (2050) self-sufficiency ratio (SSR) of rice in each country.** The SSR is calculated as the ratio of rice production to domestic rice consumption. The countries included are Burkina Faso (BFA), Côte d'Ivoire (CIV), Egypt (EGY), Ethiopia (ETH), Ghana (GHA), Kenya (KEN), Madagascar (MDG), Mali (MLI), Niger (NER), Nigeria (NGA), Rwanda (RWA), Senegal (SEN), Tanzania (TZA), Uganda (UGA), and Zambia (ZMB). The graph is sorted in descending order of the current SSR. The bars in green and brown represent the current (2018-2020) and future (2050) SSR, respectively. Insets show bar charts for the (**A**) current (green) and future (brown) SSR for Africa and (**B**) future rice deficit (orange) in each country. The rice deficit is calculated as the difference between projected rice demand and extrapolated rice production by 2050. The future SSR and rice deficit were estimated by assuming a continuation of the historical yield trend under the current rice area in each country, see Methods. Source data are provided as a Source Data file.

exhibited a considerable exploitable yield gap (>30% of attainable yield), except for irrigated rice in Egypt and Senegal where farmers have nearly closed the exploitable yield gap. Water regime and environment influenced the magnitude of exploitable yield gaps, with gaps being smaller in irrigated rice (47% of attainable yield) compared with rainfed lowland rice and upland rice (53% and 58% of attainable yield, respectively). Still, irrigated rice and rainfed lowland rice offer the best opportunity for increasing regional production given their higher absolute exploitable yield gaps (3.7 and 5.7 Mg ha⁻¹, respectively, for irrigated rice and rainfed lowland rice) compared with upland rice (2.4 Mg ha⁻¹) and higher stability as quantified using the inter-annual coefficient of variation in yield potential (Supplementary Fig. 3). Our study also identified regions with largest room for increasing yield at regional, national, and subnational levels. For example, while Egypt in North Africa and Senegal in West Africa achieved yields close to yield potential, East Africa exhibited the largest exploitable yield gap for irrigated rice within Africa, whereas large gaps were common across the entire area cultivated with rainfed rice (Fig. 2).

## Future rice self-sufficiency and rice imports

None of the 15 countries included in our analysis were self-sufficient for rice at present, with the self-sufficiency ratio (SSR) ranging from ca. 0.85 (Mali and Tanzania) to less than 0.20 (Ethiopia, Kenya, and Niger) (Fig. 3). Considering all the 15 countries included in our analysis, current area-weighted average SSR was 0.67, ranging from 0.14 to 0.89. When considering the entire African continent, average SSR becomes smaller (0.57) due to inclusion of other countries that consume rice but produce little, as it is the case for countries in North Africa (Supplementary Fig. 4). Doubling population in Africa over the next 30 years (from 1.3 to 2.5 billion), together with greater rice consumption per capita (from 48 to 60 kg per capita), would lead to 135% increase in demand for rice, totaling 150 Mt by 2050 (Supplementary Table 5).

Continuation of the historical rate of yield trend, without cropland expansion, would lead to substantially lower rice SSR by year 2050 (Fig. 3). For example, SSR by year 2050 will range from 0.03 in Kenya to 0.47 in Tanzania, averaging 0.26 for the whole continent (Fig. 3). Besides SSR, it is pertinent to know the annual rice deficit,

defined as the difference between projected rice demand and production by year 2050, as it determines the absolute requirement of imports and/or extra crop area that would be needed to meet domestic demand (Fig. 3). We found that the rice deficit will sum up to 67 Mt by 2050, which is equivalent to 20 billion US$ of rice imports at current prices or 23 M ha of new rice area at current yield level. Also relevant is to look at the rice deficit across countries to identify cases where increasing current production is more pressing. For example, we found that the rice deficit will be over 6 Mt in Côte d'Ivoire, Mali, and Madagascar, and as high as 16 Mt in Nigeria (Fig. 3). Thus, without substantial increases in crop yields, the region will experience a massive increase in rice imports and/or land conversion into rice production.

We next assessed requirements for rice imports and cropland expansion by year 2050 for different scenarios of crop intensification (Fig. 4). For our assessment, we constrained cropland expansion between 0.2 and 0.6 M ha per year, which are equivalent to ±50% of the rice area expansion rate over the past 30 years (0.4 M ha per year). We also assumed that there are no changes in the fraction of irrigated area and cropping intensity and that regional trade can offset rice deficits in specific countries through the surplus generated in other countries within the continent. We found that crop intensification can drastically reduce land requirements and need for imports. Given current rate of cropland expansion (0.4 M ha per year), a full closure of the current exploitable yield gap would eliminate the need for rice imports by year 2050. However, full closure of the exploitable yield gap would require a regional rate of yield gain of 104 kg per ha per year. Sustaining such high rate of yield gain over the next 30 years seems difficult, considering that average yields in Africa have not increased over the past 30 years (Fig. 1). Perhaps a more realistic scenario would be one with half closure of current exploitable yield gap, requiring yield gain rates (average: 52 kg per ha per year) that are comparable to those observed during the Green Revolution in Asia and elsewhere and, thus, feasible to achieve in Africa through improvements in management practices[6,31,32], and current rice area expansion rate (0.4 M ha per year), which would lead to an additional area of ca. 12 M ha of rice by 2050. In this scenario, SSR will increase from 0.57 (current) to 0.82 (2050),

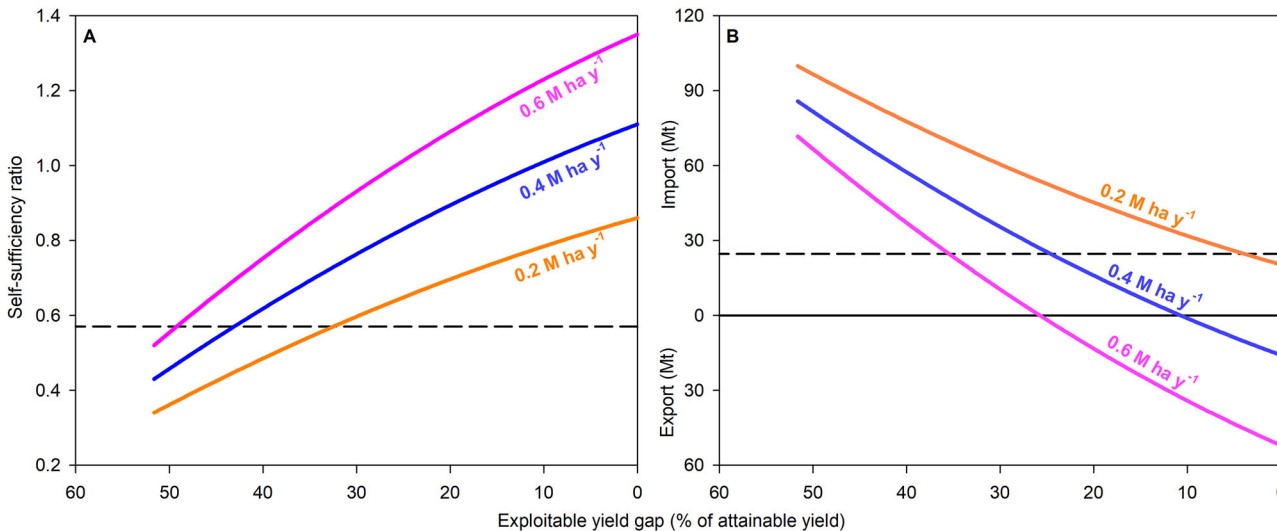

**Fig. 4 | Assessment of rice self-sufficiency in Africa by year 2050 under different scenarios of rice yield improvement and rice area expansion. A** Rice self-sufficiency ratio calculated as the ratio between rice production and rice consumption. **B** Rice import requirement, calculated as the difference between rice consumption and rice production. Intensification range goes from 52% (current) to 100% closure of the exploitable yield gap while three scenarios of rice area expansion are shown: current, faster, and slower (0.4, 0.6, and 0.2 M ha per year, respectively). Horizontal dashed line represents current rice self-sufficiency ratio (**A**) and current rice import (**B**) in Africa (2018-2020 average), whereas the solid line represents full rice self-sufficiency. Source data are provided as a Source Data file.

more than tripling the current total rice production (Fig. 4; Supplementary Fig. 5). Despite higher SSR, rice imports remain at nearly the same levels as today because of the higher absolute demand by 2050 compared with the current baseline.

In another scenario, halving the current exploitable yield gap together with faster rice area expansion (0.6 M ha per year) can lead to rice self-sufficiency in Africa by 2050 (Fig. 4). However, such scenario would require more than doubling rice area over the next 30 years (from current 15 M ha to 33 M ha by year 2050), which may be difficult to achieve due to required investments (infrastructure, irrigation schemes, roads, etc.), limited availability of suitable land for rice cultivation in current and future climate scenarios, and international pressure to reduce conversion of natural ecosystems and methane emissions from rice[33–35]. On the other hand, we note that slowing or eliminating rice area expansion has a detrimental impact on rice self-sufficiency and imports requirements. For example, if current rate of rice area expansion is reduced by half over the next 30 years (from 0.4 to 0.2 M ha per year), narrowing the exploitable yield gap by half will not be sufficient to maintain current import levels, and the region's dependance on imports will increase two-fold, at a total annual cost of 16 billion US$ by year 2050 (Supplementary Fig. 6). Thus, a combination of yield gap closure at rates that are similar to those observed during the Green Revolution in Asia so that the exploitable yield gap is narrowed by half, together with the continuation of current trajectories in rice areas, seems to be the best compromise to achieve a reasonable level of rice self-sufficiency in Africa, while avoiding a huge increase in imports and acceleration of land conversion for rice cultivation.

## Discussion

Producing sufficient food to meet global food demand by 2050, without massive land conversion for agriculture, would be a significant challenge[36]. This is particularly critical in regions with projected high population increase and limited monetary reserves to afford food imports, as it is the case of most African countries[37]. One may argue that it is crucial to consider the overall self-sufficiency of cereals, rather than focusing on individual crops. However, densely populated nations implement policies aimed at achieving self-sufficiency in key staple crops, with the aim of averting disruptions in the supply chain

that could threaten food security[12]. In the case of Africa, rice demand is increasing at a faster pace than any other food staple due to population growth, rising incomes, and a shift in consumer preferences in favor of rice[2,3,38]. Whereas substitution of rice by other crops is theoretically possible, we note that (i) rice demand has remained stable during the recurrent food crises affecting the region[5–7] and (ii) other major staple crops in the region also exhibit large yield gaps, which limits the degree to which they can buffer rice shortages[10]. In this context, it is desirable for African nations to implement policies aimed to achieve a reasonable level of self-sufficiency for rice[10,39,40]. Our study suggests that Africa will need a combination of yield intensification, similar to that occurring during the Green Revolution in Asia, together with modest area expansion, to improve SSR and avoid further increase in rice imports and associated costs. While the strategy for increasing rice production must be tailored to each country, the overall message is that Africa must raise current yields substantially to avoid the need for larger amount of rice imports and massive expansion of cropland. Failure to do so could have far-reaching consequences, such as aggravating the food insecurity and political turmoil that already exists in Africa[41,42], in the context of climate change, global rice price volatility, and trade restrictions in reaction to tight supply situations[42–46]. Our analysis also emphasizes the importance of maintaining a large rice surplus in regions that are now net exporters, such as Southeast Asia and North and South America, to be able to supply rice to Africa[31].

We find that the regional area-weighted average yield potential for Africa (8 Mg ha⁻¹ per crop) is lower and less stable than that reported for other rice-producing areas located in tropical and subtropical areas such as Southeast Asia, where the yield potential is ca. 9 Mg per ha per crop[31]. These differences are related to the larger share of rice area accounted by rainfed upland and lowland rice in Africa[19]. However, differences in yield potential between Africa and Southeast Asia were comparably smaller to those in yield gaps. We find that the average exploitable yield gap represents nearly half of the attainable yield in Africa. For comparison purposes, the exploitable yield gap represents 36% and 14% of the attainable yield in Southeast Asia and China, respectively[19,31]. Yield gaps for Africa reported here are within the range of those published in previous studies for specific countries[10,11,25,47]. Our study makes an important contribution at mapping yield gaps for most of the rice area in Africa, separately for each

country, environment, and water regime, providing a basis for prioritizing agricultural R&D and investments at regional, national, and subnational levels aiming at improving rice self-sufficiency by 2050. For example, our study revealed a large exploitable yield gap (>70% of attainable yield) in nearly one-third of the existing rice-producing area. Causes for yield gaps can be attributed to several factors, including soil water status and water management, soil fertility and fertilizer management, weed control, varietal choice and seed source, and crop establishment method[24,47,48]. Among these factors, insufficient nutrient supply seems to be key factor in explaining yield gaps for a given water regime[49,50]. For example, nitrogen fertilizer in irrigated rice production in Sub-Saharan Africa is considerably lower than those used in irrigated rice in Egypt and Senegal and elsewhere to produce yield that reach ca. 80% of yield potential[19,51]. Likewise, risk management plays a crucial role for rainfed rice production, where uncertainty on water supply imposes another constraint to adoption of agricultural inputs such as fertilizers and pesticides[27]. Furthermore, the socioeconomic factors such as limited access to knowledge, finance, inputs, and markets have significantly impeded farmers' capacity to close yield gaps, in parallel with a context of limited investments in agricultural R&D, political turmoil, and conflicts within and among countries[9,42,52]. However, several measures can address this issue, including strengthening extension services, providing fertilizer subsidies, introducing credit guarantee schemes, and establishing mechanisms to stabilize prices of agricultural inputs. To summarize, considerable room for rice yield improvement exists, but achieving it would require an unprecedented investment on national and regional agricultural R&D programs with an explicit focus on yield intensification.

Even when the yield gap is substantially closed, our analysis shows that expanding rice area in Africa will be still needed to avoid a substantial increase in rice imports by 2050 (Fig. 4). However, land requirements will depend on where rice area expands and its associated productivity level. For example, fostering further expansion of upland rice area, which currently accounts for one third of Africa's rice area, does not seem a sound strategy to satisfy future demand for rice considering its low productivity and stability[53,54]. Given the ample availability of suitable land and water resources for irrigated rice cultivation and little competition with upland crops in lowland environments[55,56], an alternative would be to allow expansion of irrigated rice area[11], which currently account for only 3.8 M ha (25% of African rice area)[4,57], while simultaneously fostering a progressive conversion of rainfed lowland rice into irrigated systems. Such approach would lead not only to higher productivity of individual crops but also allow higher cropping intensity, ultimately reducing land requirements[11]. Indeed, we note that rice cropping intensity is low in Africa (1-2 crops per year) compared to Southeast Asia (2-3 crops per year), suggesting room for increasing the overall productivity of the cropping system through greater cropping intensity[31] (Supplementary Fig. 1). On the other hand, we are cautious about the scope for increasing rice intensity given the prevalence of rice-vegetable systems in Sub-Saharan Africa, which supports our assumption of no changes in crop intensity for our scenario assessment[32,58,59]. As we mentioned previously, expanding cropland could have a substantial environmental impact due to conversion of natural ecosystems for rice production and higher greenhouse gas emissions associated with flooded rice[17,33–35]. Also, we note that there could be negative impacts of climate change on land suitability for rice and availability of water for irrigation, together with growing concern on unsustainable water extraction linked to expansion of irrigated rice area[33,34,60]. On the other hand, if expansion of irrigated area is constrained, it will put further pressure to expand rice area in low-yield, high-risk environments, ultimately leading to larger land conversion and reliance on imports. Thus, improving yields through judicious and environmentally sound inputs of fertilizer and pesticides and improved crop and soil practices, increasing crop intensity and the irrigated area fraction wherever

possible, and some expansion of rice cultivation into areas with lower environmental impact are the most realistic options to increase regional rice production in the short term[9,19,39]. Ultimately, a combination of these strategies, supported via investments in agricultural R&D programs and proper policy, is the most viable pathway to ensuring a reasonable level of rice self-sufficiency in the region, while minimizing negative environmental impact[24,61]. These strategies should give priority to identify most suitable land for further rice area expansion, rehabilitate irrigation schemes where necessary, increase availability and access to agricultural inputs (improved seeds, fertilizers, pesticides, and farm machinery), and improve the capacity for technology development, training, and dissemination[9,11,46,62].

Our analysis is subjected to several uncertainties. The weather, soil, management, and average yield data used as input for simulating crop models, as well as that used for calibration of the model, would impact the magnitude of the yield potential and yield gaps. To minimize this uncertainty, we followed the protocols of the Global Yield Gap Atlas, which give preference to best available sources of weather, soil, management, and yield data. More importantly, our crossvalidation showed that our estimated yield potential is consistent with independent estimates of yield potential derived from well-managed crops in field trials and with the yield potential estimated for similar climate zones in other rice-producing regions of the world (Supplementary Table 4). Our study did not consider the negative impact of climate change on crop yield, either directly through changes in temperature and precipitation or indirectly via pest and disease outbreaks[35], as well as on the land suitability for rice production[33]. However, we note that its impact on our assessment for 2050 can be considered minor given the relatively small change in climate projected for the first half of the century and is likely to be offset by changes in rice varieties and crop management practices together with $CO_2$ fertilization[63–65]. Ultimately, any negative impact of climate change on yield potential and/or land suitability for rice production over the next 30 years will add further pressure to increase current rice yields in Africa as a pathway to increase production, while reducing requirements for extra rice area and associated methane emissions. Overall, the expected effects of climate change on yield and land suitability for rice in Africa during the first half of the century (from 5% to 10%)[57,66–68], will have a relatively minor impact on outcomes of our scenario assessment as shown via sensitivity analysis (Supplementary Fig. 7). Likewise, our scenario analysis does not incorporate the potential improvement in rice yield potential resulting from genetic enhancements or adoption of hybrid rice. However, limited progress has been achieved on improving yield potential for rice varieties and the yield increase realized from hybrid varieties is much smaller than the size of the current yield gap and there are considerable barriers for the adoption of hybrid rice at scale[69,70]. In any case, the size of yield change due to climate change and/or genetic improvement seems much smaller than the current yield gaps, highlighting the room that exists to increase average farmer yield via improved management practices. Finally, we note that achieving the desired degree of crop intensification in the socio-economic context of Africa is challenging. For example, our assumption that the yield gap can be narrowed in 30 years may be overoptimistic. However, we note that rice smallholders in Africa actively participate in markets, as it is the case in Southeast Asia, India, and China, rather than solely relying on subsistence farming[9,13], and there is empirical evidence from other rice-producing countries in Asia, and even in Africa (e.g., Senegal, Egypt), showing that closing the current yield gap by 50%, which is taken as the more realistic yield intensification scenario in our study, is possible within a relatively short timeline via an explicit investment on agricultural R&D programs to foster rice intensification. Hence, it seems appropriate to use this level of yield gap closure to assess to which degree rice production in Africa could potentially be increased if there is a conscious investment on agricultural R&D programs and policies oriented

towards rice intensification in the region. Ultimately, our assessment demonstrates the urgent need for yield improvement in rice systems in Africa to achieve a reasonable level of self-sufficiency and reduce associated land and/or import requirements.

## Methods
### Site selection
Our study focused on the 15 rice-producing countries in Africa, including Egypt in North Africa, Burkina Faso, Côte d'Ivoire, Ghana, Mali, Niger, Nigeria, and Senegal in West Africa, and Ethiopia, Kenya, Madagascar, Rwanda, Tanzania, Uganda, and Zambia in East Africa[71] (Supplementary Information Text Section 1 and Supplementary Fig. 1 and Table 1). These countries account for 65% and 80%, respectively, of the total harvested rice area and production in Africa (average from 2018-2020)[6]. The 15 countries portray the diversity of rice cropping systems and agro-ecological zones where rice is grown across Africa. We focused on the three primary types of rice production environment in Africa: irrigated rice, rainfed lowland rice, and rainfed upland rice production[27] (Supplementary Tables 2 and 3). We noted that due to the small rainfed rice area, only irrigated rice was considered in Egypt, Kenya, Madagascar, Niger, and Rwanda. Similarly, only rainfed rice (lowland and/or upland) was considered in Côte d'Ivoire, Ethiopia, Uganda, and Zambia. Both irrigated and rainfed rice were considered in the remaining six countries: Burkina Faso, Ghana, Mali, Nigeria, Senegal, and Tanzania[4,24,57,72] (Supplementary Information Text Section 2 and Supplementary Table 1). Overall, we included a total of 20 country-water regime combinations in our study.

We selected a number of representative sites following the Global Yield Gap Atlas (GYGA) protocol (www.yieldgap.org)[28,73,74]. Briefly, the Spatial Production Allocation Model map (SPAM 2010; www.mapspam.info), together expert knowledge from colleagues from Africa Rice Center (www.africarice.org) and national partners, were used to identify the distribution of the rice harvested area in each of the 20 different country-water regime combinations[75]. For each of the country-water regimes, we selected one or more reference weather stations (RWS) based on the current distribution of weather stations, rice harvested area, and a climate zone (CZ) scheme that accounts for spatial variation in three key parameters affecting crop yield and its variability: annual growing-degree days, aridity index, and temperature seasonality[73,74].

Following this approach, we selected the CZs where rice is grown that accounts for more than 5% of the total harvested rice area for each water regime in each country[73,74]. A buffer with a radius of 100 km was created around each RWS, and this circle was then clipped by the CZ where the RWS was located. We selected buffers for each country-water regime combination, beginning with the one that had the largest harvested rice area and following with the one with the second largest areas, after discarding buffers that overlapped with the selected buffers by more than 20%. This process was continued until the overall rice coverage across all the selected buffers reached at least 50% of the national total harvested rice area for each water regime. Additional RWS were created for those rice-producing areas where weather stations did not exist[21]. The final selected sites were checked by researchers (and corrected as needed) to ensure proper representation of rice production areas and revised as needed. At the end, we selected a total of 45, 45, and 26 RWS for irrigated rice, rainfed lowland rice, and rainfed upland rice, respectively (Supplementary Information Text Section 2 and Supplementary Fig. 2 and Table 2). Selected RWS buffers accounted for 38% of rice harvested area across the 15 countries. The limited coverage by RWS buffers was due to low coverage of rainfed rice compared with irrigated rice (28% and 55%, respectively). Achieving a higher coverage of the rainfed rice area was challenging given the spread of its harvested area and the lack of data (weather, soil, management, and yield) needed for the simulations. Nevertheless, the coverage can be considered acceptable considering that selected

sites are located in CZs that account for 54% and 71% of rainfed and irrigated rice area, altogether covering two-thirds of the total rice harvested area across these countries (Supplementary Information Text Section 2).

### Weather data
Our goal was to determine the average yield potential for each site via crop modeling. Hence, it is critical to have a reasonable number of years included in our simulations to account for the effect of year-to-year variation in weather on yield potential. Previous studies have shown that 10 years are sufficient for robust simulation of yield potential and its variations in crops grown in favorable environments, as it is the case of irrigated and rainfed lowland rice in our study[76,77]. For crops grown in unfavorable environment, as it is the case of rainfed upland rice in our study, more years (15 to 20) are needed. In the present study, we simulated yield potential (or water-limited yield potential for rainfed rice) and its year-to-year variation based on 20 years of recent weather data (2000–2019) retrieved for the RWS in 10 of the 15 countries, including Burkina Faso, Ghana, Mali, Niger, Nigeria, Ethiopia, Kenya, Tanzania, Uganda, and Zambia. For the other five countries (Egypt, Côte d'Ivoire, Senegal, Madagascar, and Rwanda), our estimates of yield potential are based on 11 years of older weather data (1995–2005). Because we did not detect any trend in yield potential over time in the 10 countries for which we had weather data from 2000 to 2019, we included the estimates of yield potential from these other five countries in our analysis. Weather data used for the simulation of yield potential in all RWS in these 15 countries included daily solar radiation, maximum and minimum temperatures, precipitation, vapor pressure deficit, and wind speed. Weather data for selected weather stations were subjected to quality control measures to fill in missing data and identify and correct erroneous values by using linear interpolation to fill out missing data (https://www.yieldgap.org/methods-weather-data). Of the total of 82 selected RWS, measured daily weather data, propagated gridded weather data[78], and gridded weather data were available for 23%, 76%, and 1% of them (Supplementary Information Text Section 4), respectively.

### Crop management
Given that yield potential depends on climate, including solar radiation, temperature, and water supply in the case of rainfed crops, it is important to simulate yield potential in the context of the current crop systems (as determined by the date and method of crop establishment, crop cycle length, and crop sequence) and environment (irrigated, rainfed lowland, and rainfed upland)[28,29]. Simulating yield potential of rice required a thorough understanding of rice-based cropping systems in Africa, including input from local experts to determine crop calendars and dominant rice environments in each country and extensive on-the-ground data collection, including weather, soil, and management information across 15 African countries that include a total of 10 M ha cultivated with rice[6,28]. Data on crop management practices for each buffer were retrieved through agronomists from AfricaRice, which is the most important rice research organization in Africa, including a vast network of researchers linked with agronomists and extension specialists, and strongly connected with policymakers in the major rice-producing countries in Africa (www.africarice.org), and national agricultural research institutes and extension agents. The requested information included dominant crop sequences, ecosystems (upland/lowland), water regime (rainfed/irrigated) and proportion of each of them to the total harvested rice area, crop establishment method (transplanted/direct-seeded), average sowing dates for both transplanted and direct-seeded rice and transplanting date for transplanted rice, and dominant rice variety name and maturity. Reported dates of establishment (either transplanting or direct seeding) correspond to the dominant establishment date of each cropping system in each region reported by local agronomists

and extension agents. Rice crop calendars for representative rice cropping systems in each country are shown in Supplementary Fig. 1. In each of the buffers, we identified dominant rice cropping systems, which are characterized by ecosystem, water regime, and rice cropping intensity.

## Yield potential simulation

We used the well-validated crop simulation model ORYZA v3 and site-specific data on weather, soil, and crop management practices to estimate yield potential. The model was developed to simulate the growth and development of rice and has been validated and used across a wide range of rice cropping systems[20,79,80]. Given the lack of experimental data from well-managed experiments needed to calibrate rice varieties, we used generic crop parameters derived for rice varieties in Africa in previously published studies[57,81–83]. Briefly, in these previous studies, an ORYZA model version named as ORYZA2000v2n14, which builds upon the ORYZA2000v2n13s14 version, was used to derive the genetic parameters of these rice varieties through iterating calibration and validation processes with initial values of crop parameters from a widely cultivated variety, IR72. This updated iteration includes enhancements in modeling heat sterility, cold sterility, and phenology[57,83]. Specifically, ORYZA2000v2n14 incorporates features such as explicit simulation of transpirational cooling and earlier flowering in hotter climates.

In each RWS buffer, we collected data on water regime, crop establishment, rice variety name, sowing or transplanting (for transplanted rice only) date, and maturity date from local agricultural specialists. Subsequently, we employed the DRATE v2 program, which was integrated into the ORYZA v3 model, to calibrate phenology development rate parameters, including development rates for juvenile (DVRJ), photoperiod sensitivity (DVRI), panicle development (DVRP), and reproductive phases (DVRR), based on data on phenological stages and growth duration[80], assuming that the 50% flowering date was fixed to occur 30 days before the maturity date[57]. The data for other variables (e.g., assuming a base temperature for development of 14 °C, a maximum optimum photoperiod of 10 h, no photoperiod sensitivity, a lower air temperature threshold for growth of 12 °C, consecutive number of days below the lower air temperature threshold that crop dies of 3 d, a critical temperature of spikelet sterility of 35.6 °C, a fraction of sunlight energy that is photosynthetically active of 0.5, and a fraction of carbohydrates allocation to stems that is stored as reserves of 0.2) were obtained from previous studies[57,81,84].

For each of the 20 country-water regime combinations, we simulated yield potential of irrigated rice (or water-limited yield potential of rainfed lowland and upland rice) for each rice cycle within the dominant cropping system (Fig. 2). We assumed no water limitation for irrigated rice, whereas simulations of rainfed lowland and upland rice accounted for the amount and distribution of precipitation and soil properties influencing the soil water balance. Water-limited yield potential simulations of rainfed rice were conducted using the assumption of a non-puddled clayey loam soil with a bund height of 25 cm for rainfed lowland rice. The effect of groundwater depth on rainfed rice yield is highly contextual, varying greatly among locations, seasons, and landscapes[57,85]. We simulated water-limited yield potential for rainfed lowland rice under two scenarios of groundwater depth to account for the variety of scenarios and associated uncertainty during the entire crop cycle (shallow [40 cm] and deep [100 cm]), as upstream and downstream valley bottom groundwater depths were in this range[86]. We also assumed that the area of rainfed lowland rice in each buffer is split evenly (50:50) between the two groundwater scenarios (40 & 100 cm deep lowland groundwater depths). For rainfed lowland rice, the two scenarios basically portray rainfed favorable (shallow water table) and drought-prone (deep water table) environments. Initial volumetric water content, saturated volumetric water content of ripened, saturated hydraulic conductivity of soil was

assumed to be $0.57\,m^3\,m^{-3}$, $0.56\,m^3\,m^{-3}$, $10.79\,cm\,d^{-1}$ for rainfed lowland rice, respectively, and $0.39\,m^3\,m^{-3}$, $0.38\,m^3\,m^{-3}$, $99.77\,cm\,d^{-1}$ for rainfed upland rice. In the case of rainfed upland rice, water-limited yield potential was simulated with a groundwater depth during the entire crop cycle of 1000 cm in a nonpuddled sandy loam soil without a bund (Supplementary Information Text Section 3). Following previous studies, sensitivity analysis was performed to assess the sensitivity of simulated yields to assumptions on parameters related to soil water holding capacity, presence of hardpan, bunding height, and groundwater table depth[81].

Validating our estimates of yield potential in Africa is challenging due to the limited availability of experimental data collected from well-managed crops that grow without nutrient limitations and kept free of weeds, diseases, and insect pests[57]. We performed a cross-validation of simulated yield potential for a subset of sites for which measured yield in well-managed rice crops were available (Supplementary Table 4). In areas lacking experimental yield data, we extended the cross-validation by comparing yield potential in a particular climate zone against that reported by the Global Yield Gap Atlas for the same climate zone in other regions of the world, such as Southeast Asia, where the yield potential has been well validated. The year-to-year variation in yield potential of irrigated rice (or water-limited yield potential of rainfed rice) was assessed by determining the inter-annual coefficient of variation and semi-deviation for irrigated versus rainfed rice and rainfed lowland versus rainfed upland rice (Supplementary Information Text Section 3 and Supplementary Table 3). The computation of semi-deviation was performed with a downside risk approach using the "PerformanceAnalytics" package in R software version 4.1.2[87].

## Yield gap estimation

For each of the 20 country-water regime combinations, the yield gap was determined for each cycle of rice production by difference between the yield potential (irrigated rice) or water-limited yield potential (rainfed lowland and upland rice) and the average yield of the farmers[29]. We note that while the yield potential for five countries, including Egypt, Côte d'Ivoire, Senegal, Madagascar, and Rwanda, were based on the average between 1995 and 2005, we used the most up-to-date data to estimate the actual farmer yields for these five countries, as well as for the other 10 countries. Data on average farmers' yields for irrigated and rainfed rice were collected separately from national statistics, previous publications and databases, and local agronomists (Supplementary Information Text Section 5 and Supplementary Table 6). All yield data reported in our study is reported as paddy rice at a standard moisture content of $140\,g\,H_2O\,kg^{-1}$ grain. The yield gap for irrigated and rainfed rice in each RWS buffer was estimated separately for each country. Average yield gap in each RWS buffer was estimated by weighting yield potential and actual farmers' yield based on the proportion of harvested rice area of each cycle in each cropping system (Supplementary Fig. 1).

## Current (2018-2020) and future (2050) rice demand and self-sufficiency

We considered the average annual domestic rice demand during the 2018–2020 period as the baseline for our study (Fig. 1). Current national domestic rice demand for each country was estimated based on the average annual national rice production, imports, exports, and stock change during the period from 2018 to 2020[5] (Supplementary Table 5). We estimated future demand for rice by 2050 in each country by multiplying their projected populations, based on the medium fertility variant of the UN population prospects[3], and the per-capita rice demand in 2050. The latter was calculated based on the relative change in average per-capita rice demand between the baseline period (2018–2020) and the year 2050, which was derived for each country from the International Model for Policy Analysis of Agricultural Commodities and Trade (IMPACT) database[8] (Supplementary Table 5). The

IMPACT projections account for various socio-economic factors, including effective consumer prices, region-specific population dynamics, income growth rates, government blending mandates, energy prices, producer subsidy equivalents encompassing subsidies and trade measures, commodity-specific indices for all commodities, as well as price and income elasticity considerations[8]. For all the 15 countries, the total rice demand that is projected for the year 2050 is predicted to be higher than the current demand (2018–2020), with the size of increase ranging from 79% to 469% (Supplementary Table 5). The increasing demand for rice is driven by both projected increase in population and per-capita rice consumption in Kenya, Madagascar, Rwanda, and Zambia, whereas in the rest of the countries, demand increase is being driven primarily by the projected increase in population. In our study, all rice yield, production, per-capita rice demand, and total rice demand were reported as paddy rice at a standard moisture content of $140\,g\,H_2O\,kg^{-1}$ rice grain. The per-capita rice demand was converted to paddy rice by dividing initially reported milled rice from the USDA database by the respective country's rice milling rate[5,8], ranging from 0.63 to 0.69 across countries (Supplementary Table 5).

To help identify hotspots for yield intensification and/or area expansion on a country basis, we calculated current and future (year 2050) SSR and rice deficit in each of the 15 selected countries (Fig. 3). The SSR was calculated for each country by dividing the annual rice production by the annual rice demand, while net import or exports was determined by subtracting the annual rice production from the annual rice demand. The rice deficit was calculated as the difference between projected rice demand and extrapolated rice production, which was estimated as the product of extrapolated rice yield by 2050 and current rice area. To extrapolate rice yield by 2050, we assumed a continuation of the historical yield increase rate observed during the past three decades in each country, until the rice yield reaches the attainable yield, estimated as 70-80% of the simulated yield potential (see next section), which was the case of Côte d'Ivoire and Uganda (Supplementary Table 7).

### Scenario assessment
In our study, we determined rice production and imports requirement in Africa for different scenarios of yield intensification and rice area expansion (Fig. 4; Supplementary Figs. 5 and 6). Following previous studies, the exploitable yield gap was defined as the difference between 80% of yield potential (irrigated rice) or 70% of water-limited yield potential (rainfed rice) and the current average farmer yield[19,88,89]. Following prior assessments on food supply-demand scenarios[10,43,90], we selected 2050 as the target year for our evaluation. This 30-year timeframe strikes a balance between minimizing the long-term impact of climate change on rice yields and cropping systems[63,66], while providing enough time to plan for structural changes, implement technologies, mitigate the risk of unpredictable events (such as economic downturns in any given year), and formulate short- and long-term policies and orient agricultural R&D programs to eliminate the exploitable yield gap.

In the case of yield intensification, we estimated rice yield under different levels of exploitable yield gap, ranging from 52% (current) to full closure of exploitable yield gap. For the purpose of area expansion assessment, we considered three scenarios for rice area expansion by 2050: (i) expanding harvested rice area by 0.4 M ha annually, which is the historical rice area expansion rate in Africa over the past three decades, (ii) expanding harvested rice area at an annual rate of 0.2 M ha, representing a 50% reduction in the historical expansion rate, and (iii) expanding harvested rice area at an annual rate of 0.6 M ha, equivalent to a 50% increase in the historical expansion rate. Note that in our analysis, we assumed that there would be no change in the fraction of irrigated rice area or changes on crop intensity.

Our scenario assessment focused on estimating the aggregated rice SSR, net rice import, and financial expenditure associated with importing rice in Africa for the current baseline and different scenarios of yield intensification and area expansion by 2050. In our scenario assessment, we considered all 58 countries and disputed territories in Africa[71]. Unfortunately, the data needed to estimate rice demand by 2050 was not available for 21 countries and disputed territories. Since these countries and disputed territories account for 5% of the current rice demand estimated for the other 37 countries, we simply multiplied the predicted annual rice demand from the 37 countries by 1.05 to determine the total annual rice demand in Africa by 2050 (Supplementary Table 5). In the case of estimation of total rice production for Africa, our study included 15 major rice producing countries, which altogether account for 80% of total rice production in Africa. For the remaining 43 countries and disputed territories that were not included in our analysis, we assumed the average yield gap derived from the 15 countries, only considering those countries that produce rice (Supplementary Fig. 5). To determine the total cost associated with rice import, we multiplied the volume of rice imported by Africa with the rice market price. Similarly, we calculated the money earned from export (surplus) by multiplying the annual rice export by the rice market price. The rice market price ($US\$289\,Mg^{-1}$ paddy rice) was derived from the World Bank annual price of rice (Thai 5%) between 2018 and 2020[7] (Supplementary Fig. 6).

Finally, a sensitivity analysis was conducted to understand the potential impact of climate change on the outcomes from our scenario assessment. Previous studies have reported 5–10% reduction in yield and land suitability for rice in Africa due to climate change[57,66–68]. Hence, we re-estimated rice production and SSR by year 2050 for each of our yield intensification scenarios under current area expansion rate for different combinations of yield potential reduction (-5% and −10%) and limited cropland expansion (-5% and −10%) (Supplementary Information Text Section 6 and Supplementary Fig. 7).

## Data availability
Data on rice yield potential from the Global Yield Gap Atlas (GYGA) are available at www.yieldgap.org. Data on rice yield, harvested area, production, export, and import from the FAOSTAT are available at www.fao.org/faostat. Data on rice distribution from the SPAM map are available at www.mapspam.info. Data on the current and future population size from the United Nations are available at https://population.un.org/wpp/. Data on rice market price from the World Bank are available at https://www.worldbank.org/en/research/commodity-markets. Data on the current per-capita rice demand and rice milling rate from the USDA database are available at https://apps.fas.usda.gov/psdonline/app/index.html#/app/advQuery. Source data are provided with this paper.

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

## Acknowledgements

We would like to thank local agronomists and extension agents in each country for their help in data collection: Adda, C., Ahounanton, K., Diagne, A., Cissé, B., El-Namaky, R., Johnson, J. M., Shrestha, S., Traoré, K., Senthilkumar, K. & Toure, A. (AfricaRice); Asai, H. & Tsujimoto, Y. (JIRCAS); Kasuya, M., Kurihara, K., Tokida, K., Tomitaka, M., Tsuboi, T. & Matsumoto, S. (JICA); Nakano, Y. (University of Tsukuba); Sedga, Z. (INERA); Mossi Maïga, I. (INRAN); Bam, R. K. (CRI); Rabeson, R. (FOFIFA); Dogbe, W. (SARI); Kajiru, G. J. (Ministry of Agriculture, Food Security); Nanfumba, D. (NARO); and Bakare, O. S. (NCRI). We acknowledge the Bill & Melinda Gates Foundation (BMGF), Seattle, USA (Grant ID INV-005431 to P.G. and M.K.v.i.) to support this project through the CGIAR Excellence in Agronomy 2030 (Incubation Phase). The author also want to thank the support from the National Key Research and Development Program of China (2022YFD2301003 to S.Y.), the Young Elite Scientists Sponsorship Program by CAST (2022QNRC001 to S.Y.), the Fundamental Research Funds for the Central Universities (2662023ZKPY004 to S.Y.), the Natural Science Foundation of Hubei Province of China

(2022CFB271 to S.Y.), the Program of Introducing Talents of Discipline to Universities in China (the 111 Project no. B14032 to S.P.), and the Belt and Road Center for Sustainable Rice Production.

## Author contributions

S.Y., K.S. and P.G. conceived and designed the study. K.S., P.A.J.v.O. and M.K.v.I. provided and compiled the data analyzed in this study. S.Y., S.B.P. and P.G. performed the spatial analysis, simulation, and data analysis. S.Y. and P.G. wrote the paper, with contribution from all authors. All authors reviewed and edited the manuscript.

## Competing interests

The authors declare no competing interests.
