## [Peer Review File · Nature Communications]

Intensifying rice production to reduce imports and land conversion in AfricaReviewer #1 (Remarks to the Author):

The paper's results are worthy to be published in Nature Communications. the methodology is solid and the authors provided enough detail in the methods for the work to be reproduced.

Besides, the results are significant and strongly match the 2° and 12° Sustainable Development Goals (SDGs) adopted by ONU indicating some of the challenges that will have to be overcome in order to meet future rice demand that will require larger rice imports and/or land conversion than now in Africa.

The paper's results can help decision-makers to accelerate action and overcome impediments that stand in the way of progress on sustainable development in Africa through science. The data analysis, interpretation, and conclusions are enough clear.

The work supports the conclusions not being needed for additional evidence. I recommend this paper for publication.

Reviewer #2 (Remarks to the Author):

Revision on:

Fostering rice intensification to reduce imports and land conversion in Africa
Shen Yuan 1, Kazuki Saito 2, Pepijn A. J. van Oort 3, Martin K. van Ittersum 4, Shaobing Peng 1, Patricio Grassini 5,*

The authors aim to assess options of increasing rice production across some countries of Africa, from increases in yield and or increases in crop land by the year 2050. The article is very well written, and the problem is important.

This article is associated to the Global Yield Gap Atlas and the information produced here, appears to already be available at <https://www.yieldgap.org/>

Issues include a significant concern, several significant assumptions made, and then there is the question about novelty.

A concern: The analysis is a commodity-based assessment (only rice) that ignores that food systems result from cumulative effects and synergies, that induce non-linear processes in multiple directions. Making very difficult to simply project demand and self-sufficiency levels linearly as done here. Involved drivers might include the substitution of foods and commodities that are more drought tolerant than rice, and not only in response to population growth as argued here. But in response to environmental, market, food policy, environmental, supply chains and logistics, infrastructure, climate change, and socio-economic factors. To note is that not even climate projections on crop productivity, land suitability, availability of water for irrigation to 2050, have been taken into consideration, not to mention changes in the frequency and intensity of extreme climate events, or associated pest and disease outbreaks. Even assuming that some of the effects might be relatively small by 2050, there is no inference on what might happen in during the second half of the century based on alternative mitigation scenarios.

It might have helped if a foresighting analysis would have been run, this is - a systematic quantitative assessment of the middle and long-term futures of science, technology, the economy, and the society - could have helped account for some of the shortcoming mentioned above and their consequences. Though such an exercise would have required to collect data and interact with a wide range of stakeholders across all these countries, which might or not might have been feasible. Instead, it is to

note that this article is just a desk-top study with rather important and to a great deal, poorly posed / explained assumptions.

Assumptions:

1. The idea of reducing targets to 70-80% of attainable yields, values commonly used in high income countries, to address risk aversion in Sub Saharan Africa smallholder farming result inappropriate. This is particularly as smallholder farming is subsistence farming not commercial farming, and farmers tend to be content with having enough rather than embarking in highly risky high input production systems as proposed here. I think this value of 80-70% needs to be better posed / explained.
2. In line 211 a required yield gain rate of 52 kg/ha/year is proposed, extrapolated from the Green Revolution in Asia – never mind that the green revolution across South East Asia caused highly significant pollution of water ways across most countries, latter associated to the subsequent slowdown in yield gains across the region (Pingali, 2012). Still assuming a 52 kg/ha/year sounds highly unrealistic given that yield gains since the 1990 (Fig 2) are close to nill. This, even though across SSA there has been continuous investment aiming to bridge yield gaps for more than 50 years (at least). Once could argue on the ineffectiveness of the CGIAR to reduce yield gaps across the continent, or the researchers might accept that the problem is a bit more complex than just adding fertilisers and new varieties. I find the discussion on the causes of yield gap rather superficial.
3. Climate change is readily missing in this report. It wouldn't have been difficult to include an analysis of sensitivity to alternative climate change projections. How about projections associated to increases in pest and disease damages in CC scenarios as reported and expected for the region in the IPCC AR6 2021.
4. The analysis of variability using coefficients of variations is rather simplistic. The authors could have used downside risk as a better measure of changes in variability. Though this might have also needed to collect additional information from stakeholders or at least assess family food demands from household surveys (all data should be available already).

Novelty:

1. I personally think that lack of novelty is a significant issue with this article. What might be useful is a food systems analysis not a commodity-based analysis. I also missed the likely benefit of alternative pathways to bridge those yield gaps. With Oryza the authors would have done an analysis to identify how that 52 kg/ha/year could be best achieved

Reviewer #3 (Remarks to the Author):

In the manuscript of 'Fostering rice intensification to reduce imports and land conversion in Africa' the authors assess the yield potential, yield gap and future rice status in Africa using the help of prediction models. The findings indicated that if yields do not dramatically rise, there will need to be a large increase in rice imports and/or land conversion in order to fulfil future rice demand. Overall, the authors' research demonstrated that the effects of climate change on rice production and the appropriateness of the land could have serious environmental consequences. Increased greenhouse gas emissions and environmental disruption could result from expanding crops. Constrained rice irrigation zones, however, might push farmers to develop in risky, low-yielding conditions, leading to more land conversion and reliance on imports. The study's impact on 2050 assessment is minor, but it will increase the need for intensifying African rice systems to reduce extra rice area requirements and methane emissions.

The methodology appears well-structured and detailed, incorporating weather data, crop simulation models, and expert knowledge to analyse yield potential, yield gaps, and future demand for rice. The selection of representative sites, consideration of various scenarios, and sensitivity analyses enhance the study's credibility. The study makes a strong case for a comprehensive strategy to solve the difficulties Africa faces in satisfying its expanding rice demand. It highlights important elements including yield potential, yield gaps, and socioeconomic limitations that have an impact on rice

production. The writers add complexity to the issue by analysing alternative remedies, including both technical and policy-based methods. Overall, the discussion part offers a clear summary of the problems relating to rice production and makes workable suggestions for solutions.

Here are specific comments and suggestions, kindly address the following:

1. Mention the methodology used in the research work in the abstract and make the abstract self-explanatory, when read it in isolation
2. Please include the previous work regarding the rice intensification studies in introduction
3. Line 80-85, point out the reasons behind the low yield despite having an increase in cultivation area,
4. Mention the land conversion pattern in Africa with respect to rice crop over the study period in the introduction part of the manuscript.
5. Fig 2 is not visible?- Some issue- Kindly check
6. Line number 112-113, Explain the process-based crop simulation models methodology you used in this study
7. Line number 45 & 118, what is ca.?
8. Line number 390-391, Generally, for climatic studies minimum of 30 years are considered. Then why only 20 years has been considered?
9. Line number 425, it will be helpful for readers if you could explain the methodology for calibration and validation of ORYZA model.
10. Crop management – How the difference in crop management has been considered
11. SSR- How it has been extrapolated to the regional level, which needs to be explained
12. Suggest to include a conclusion section, with suggestions and strategies for future directions
13. Uncertainties involved in the study and possible strategies to overcome the same
14. Improving yields through sustainable and environmentally friendly practices, utilizing existing cropland efficiently, and carefully considering the impacts of cropland expansion. Ultimately, a combination of these strategies, alongside investments in agricultural research and infrastructure, is likely the most viable path to ensuring food security while minimizing negative environmental and social consequences- So these points needs to be focused in the revision

Responses to Reviewers' Comments (and our responses **in red)**

We thank the editor and the three reviewers for their comments on our paper. We have addressed all their comments in the revised MS. Our point-by-point responses to the comments are shown below in red.

We have carefully and constructively addressed all the major points raised by reviewers. Briefly, we added text to explain the calibration of ORYZA model (L 512-514 & L 519-523 of revised MS) and assumptions (L 523-529 & L 546-549 of revised MS) and added a cross-validation of model outputs (L 137-140 & L 556-564 of revised MS, Supplementary Table 4 of revised SI). We also added text and new analyses to discuss the effect of climate change on intensification and land suitability for rice and their implications for policy intervention (L 341-342 & L 367-378 of revised MS, Supplementary Fig. 7 of revised SI) and a strong justification on why a commodity-oriented approach is appropriate to inform agricultural research and extension programs in Sub-Saharan Africa (L 265-267 & L 269-274 of revised MS). Furthermore, we added a section on uncertainties and limitations to the Discussion section to elaborate on the assumptions behind our scenario assessment (L 359-398 of revised MS).

See our detailed responses to reviewers' comments below.

Comments from Reviewer #1:

The paper's results are worthy to be published in Nature Communications. the methodology is solid and the authors provided enough detail in the methods for the work to be reproduced.

Answer: We appreciate the strong support for publication.

Besides, the results are significant and strongly match the 2° and 12° Sustainable Development Goals (SDGs) adopted by ONU indicating some of the challenges that will have to be overcome in order to meet future rice demand that will require larger rice imports and/or land conversion than now in Africa. The paper's results can help decision-makers to accelerate action and overcome impediments that stand in the way of progress on sustainable development in Africa through science. The data analysis, interpretation, and conclusions are enough clear. The work supports the conclusions not being needed for additional evidence. I recommend this paper for publication.

Answer: Thanks for highlighting the value of our study and recommendation for publication.

Comments from Reviewer #2:

The authors aim to assess options of increasing rice production across some countries of Africa, from increases in yield and or increases in crop land by the year 2050. The article is very well written, and the problem is important.

Answer: Thanks for highlighting the value of our study.

This article is associated to the Global Yield Gap Atlas and the information produced here, appears to already be available at <https://www.yieldgap.org/>.

Issues include a significant concern, several significant assumptions made, and then there is the question about novelty.

A concern: The analysis is a commodity-based assessment (only rice) that ignores that food systems result from cumulative effects and synergies, that induce non-linear processes in multiple directions. Making very difficult to simply project demand and self-sufficiency levels linearly as done here. Involved drivers might include the substitution of foods and commodities that are more drought tolerant than rice, and not only in response to population growth as argued here. But in response to environmental, market, food policy, environmental, supply chains and logistics, infrastructure, climate change, and socio-economic factors. To note is that not even climate projections on crop productivity, land suitability, availability of water for irrigation to 2050, have been taken into consideration, not to mention changes in the frequency and intensity of extreme climate events, or associated pest and disease outbreaks. Even assuming that some of the effects might be relatively small by 2050, there is no inference on what might happen in during the second half of the century based on alternative mitigation scenarios.

It might have helped if a foresighting analysis would have been run, this is - a systematic quantitative assessment of the middle and long-term futures of science, technology, the economy, and the society – could have helped account for some of the shortcoming mentioned above and their consequences. Though such an exercise would have required to collect data and interact with a wide range of stakeholders across all these countries, which might or not might have been feasible. Instead, it is to note that this article is just a desk-top study with rather important and to a great deal, poorly posed / explained assumptions.

Answer: Thanks for these comments. For simplicity, we responded separately to the three main sources of concern raised by Reviewer #2.

#1. Comment on commodity-based assessment: There are important reasons why it is necessary to look at rice from the perspective of an individual crop. First, there is a strong shift towards greater rice consumption in Sub-Saharan Africa in relation to other food crops, which will put pressure on increasing rice production or, alternatively, greater rice imports. Second, rice maintains its importance in Africa, which became clear by the stable rice consumption even during the 2008 global rice crisis, without rice being substituted by other cereal crops. Third, densely populated nations frequently implement policies targeted at achieving self-sufficiency in key staple crops, with the aim of averting disruptions in the supply chain that could threaten food security. In this context, it seems justified for Africa to seek for a reasonable level of rice self-sufficiency through investment on agricultural research and development programs and proper policy to foster rice intensification. Based on the reviewer's comment, we added further text to the MS and additional references to better justify our commodity-based assessment (L 265-267 & L 269-274 of revised MS). On a personal note, having attended many meetings with African governments and research organizations, we have the strong perception that their priority setting in agricultural R&D is strongly commodity-based - we don't recall any conversation in terms of calorie equivalents and/or policy relying on crop substitution. Thus, to be useful in providing input to research programs and policy makers, we believe that a commodity-based, rather than overall-cereal approach, is needed, especially if the goal of the Nature Communications journal is to promote good science that can lead to sound agricultural R&D prioritization.

#2. Comment on not inclusion of other factors (environmental, market, food policy, environmental, supply chains and logistics, infrastructure, climate change, and socio-economic factors): We apologize if the original text was not clear regarding future rice

demand estimates. Please, note that we are NOT projecting future rice demand and self-sufficiency linearly. Future rice demand for each country was estimated based on the projected population derived from the medium fertility variant by the 2050 (United Nations, 2022) and per-capita rice consumption by 2050 as estimated using the International Model for Policy Analysis of Agricultural Commodities and Trade (IMPACT). This model accounts for various factors when simulating future rice demand, including effective consumer prices, region-specific population dynamics, income growth rates, government blending mandates, energy prices, producer subsidy equivalents encompassing subsidies and trade measures, commodity-specific indices for all commodities, as well as price and income elasticity considerations (Robinson et al., 2015). Thus, our demand projections already accounted for most of the factors listed above by Reviewer #2. Based on this comment, we elaborated further on the explanation of our rice demand projections to make clear that we are already accounting for multiple socio-economic factors, and we are not simply extrapolating linearly into the future (L 598-602 of revised MS).

#3. Comment on climate change: As Reviewer #2 suggested above, the impact of climate change can be assumed to be relatively minor over the next 30 years. Indeed, we note that expected changes in temperature and rainfall in the main rice producing areas over the next 30 years are expected to be relatively minor (Rosenzweig et al., 2014; Challinor et al., 2014) and its negative impact on yield potential is likely to be offset by improved management practices and varietal adaptation plus CO₂ fertilization. And since our assessment focused on the next 30 years, the comment on what may happen over the second half of the century (when the impact of climate change is expected to be larger) is not relevant as we are not looking beyond 2050. Perhaps more importantly, any change in yield potential due to climate change over the next 30 years seems much smaller than the current yield gap between potential and average yield, emphasizing the room that exist to increase crop yield regardless climate change. Hence, the conclusions of the study are robust and climate change will only amplify the need to close the current yield gap as an approach for Africa to meet a reasonable fraction of their domestic demand while reducing land requirement for rice production and the associated environmental impact. As requested by the reviewer, we added text to the MS to explain better why we did not account for the impact of climate change (L 367-380 & L 384-387 of revised MS). Additionally, we performed a sensitivity analysis in which yield potential and cropland expansion were reduced by 5% and 10% to see how our findings were sensitive to possible changes due to climate change. We found that our findings are robust and, again, further emphasizing the need to close the current yield gaps. We elaborated on the sensitivity analysis in the revised MS (L 376-379 & L 669-675 of revised MS, Text Section 6 and Supplementary Fig. 7 of revised SI).

Finally, in relation to stakeholders' engagement, please note that Africa Rice Center is part of this study. Africa Rice Center (www.africarice.org) is the biggest rice research organization in Africa and includes a vast network of researchers linked with agronomists and extension workers from national programs and is well connected with policymakers in the main rice producing countries in Africa (L 487-489 of revised MS). Therefore, our assessment was based on a thorough understanding of the rice-based systems in Africa, which required a lot of ground data collection, including weather, soil, and management information across main rice producing areas in 15 countries in Africa (altogether 10 million ha) and required strong agronomic input to determine crop calendars and environments at each site. In response to the reviewer's comment, we elaborated further on how the data was collected to make sure that readers understand that this study goes far beyond a simple desktop exercise (L 101-104 & L 477-485 of revised MS).

Assumptions:

1. The idea of reducing targets to 70-80% of attainable yields, values commonly used in high income countries, to address risk aversion in Sub Saharan Africa smallholder farming result inappropriate. This is particularly as smallholder farming is subsistence farming not commercial farming, and farmers tend to be content with having enough rather than embarking in highly risky high input production systems as proposed here. I think this value of 80-70% needs to be better posed / explained.

Answer: Thanks for this comment. Attaining a yield of 70-80% of the potential is a realistic target for farmers with access to markets, resources, and extension services as reported by previous studies (Lobell et al., 2009; Yuan et al., 2021) and as it is the case of smallholder rice farmers in China, Vietnam, Indonesia, and even some African countries included in our study as it is the case of Senegal and Egypt (L 305-308 & L 628-630 of revised MS). Hence, it seems appropriate to use this level of yield gap closure to assess to which degree rice production in Africa could potentially be increased if there is a conscious investment in agricultural R&D programs and policies oriented towards rice intensification in the region. We have added text elaborating on these issues to further support the 70-80% of yield potential as a goal for future yields in Africa for a scenario in which there is a targeted investment to foster rice intensification (L 346-350 of revised MS). Having said, we agree with the reviewer that reaching 70-80% of yield potential in only 30 years seems unrealistic and that is why, once we determined the available room for increase rice production, our subsequent analysis focused on the scenario of 50% closure of current gap, which, as we said, would be a more realistic scenario given the empirical evidence from Africa and elsewhere on what is realistic to achieve via crop intensification within a relatively short timeframe (L 219-223 & L 387-395 of revised MS).

2. In line 211 a required yield gain rate of 52 kg/ha/year is proposed, extrapolated from the Green Revolution in Asia – never mind that the green revolution across South East Asia caused highly significant pollution of water ways across most countries, latter associated to the subsequent slowdown in yield gains across the region (Pingali, 2012). Still assuming a 52 kg/ha/year sounds highly unrealistic given that yield gains since the 1990 (Fig 2) are close to null. This, even though across SSA there has been continuous investment aiming to bridge yield gaps for more than 50 years (at least). One could argue on the ineffectiveness of the CGIAR to reduce yield gaps across the continent, or the researchers might accept that the problem is a bit more complex than just adding fertilisers and new varieties. I find the discussion on the causes of yield gap rather superficial.

Answer: See our previous response. Briefly, in a context of a substantial rice deficit projected for Africa by 2050, closing the existing exploitable yield gap at least by half could significantly mitigate the need for extensive rice imports and expansion of rice area. In turn, this would require an annual yield gain rate of 52 kg per ha between now and year 2050-- a rate that seems possible given the empirical evidence that exist from the Green Revolution in Asia and yield gains in other parts of the world. Also please note that the text refers to the Green Revolution in terms of the magnitude of yield gain that is needed, not necessarily the means to achieve these gains. We do not see why judicious, environmentally-sound addition of nutrients, proper pest control and water management, and use of proper varieties could not deliver similar yield gains for Africa if, again, there is an explicit initiative to foster on-farm yield improvement for rice in the continent. Based on this comment, we have rephrased the

text to enhance clarity and avoid any potential confusion (L 221-223 & L 346-357 of revised MS).

3. Climate change is readily missing in this report. It wouldn't have been difficult to include an analysis of sensitivity to alternative climate change projections. How about projections associated to increases in pest and disease damages in CC scenarios as reported and expected for the region in the IPCC AR6 2021.

Answer: See our previous response on this and the sensitivity analysis included in the revised article (L 376-380 & L 669-675 of revised MS, Text Section 6 and Supplementary Fig. 7 of revised SI). In the case of pest and diseases, yield potential is determined by climate and soil type (rainfed crops). Hence, accounting for future pressure is not relevant for estimating potential production, although it is important in determining the required interventions to increase farmer yield. Thus, we added text to acknowledge the importance of pest management under the concerns of climate change (L 367-370 of revised MS).

4. The analysis of variability using coefficients of variations is rather simplistic. The authors could have used downside risk as a better measure of changes in variability. Though this might have also needed to collect additional information from stakeholders or at least assess family food demands from household surveys (all data should be available already).

Answer: Following reviewer's suggestion, we assessed downside risk by calculating the yield semi-deviation for irrigated *versus* rainfed rice and rainfed lowland *versus* rainfed upland rice (L 130-131 & L 564-570 of revised MS and Supplementary Table 3 of revised SI).

Novelty:

1. I personally think that lack of novelty is a significant issue with this article. What might be useful is a food systems analysis not a commodity-based analysis. I also missed the likely benefit of alternative pathways to bridge those yield gaps. With *Oryza* the authors would have done an analysis to identify how that 52 kg/ha/year could be best achieved.

Answer: See our previous responses on these issues. Briefly, we added further text to the MS and additional references to better justify our commodity-based assessment (L 265-267 & L 269-274 of revised MS). In a context of a substantial rice deficit projected for Africa by 2050, closing the existing exploitable yield gap at least by half could significantly reduce the need for substantial rice import and expansion of rice area. Achieving this necessitates an annual yield gain rate of 52 kg per ha between now and year 2050-- a rate that seems possible given the empirical evidence from the Green Revolution in Asia and yield gains in other parts of the world. We also incorporated additional text to elucidate judicious and environmentally-sound addition of nutrients, proper pest control and water management, and use of proper varieties have the potential to achieve similar yield gains for Africa if there is an explicit initiative to foster on-farm yield improvement for rice (L 346-357 of revised MS).

Comments from Reviewer #3:

In the manuscript of 'Fostering rice intensification to reduce imports and land conversion in Africa' the authors assess the yield potential, yield gap and future rice status in Africa using the help of prediction models. The findings indicated that if yields do not dramatically rise, there will need to be a large increase in rice imports and/or land conversion in order to fulfil

future rice demand. Overall, the authors' research demonstrated that the effects of climate change on rice production and the appropriateness of the land could have serious environmental consequences. Increased greenhouse gas emissions and environmental disruption could result from expanding crops. Constrained rice irrigation zones, however, might push farmers to develop in risky, low-yielding conditions, leading to more land conversion and reliance on imports. The study's impact on 2050 assessment is minor, but it will increase the need for intensifying African rice systems to reduce extra rice area requirements and methane emissions.

Answer: Thanks for highlighting the value of our study.

The methodology appears well-structured and detailed, incorporating weather data, crop simulation models, and expert knowledge to analyse yield potential, yield gaps, and future demand for rice. The selection of representative sites, consideration of various scenarios, and sensitivity analyses enhance the study's credibility. The study makes a strong case for a comprehensive strategy to solve the difficulties Africa faces in satisfying its expanding rice demand. It highlights important elements including yield potential, yield gaps, and socioeconomic limitations that have an impact on rice production. The writers add complexity to the issue by analysing alternative remedies, including both technical and policy-based methods. Overall, the discussion part offers a clear summary of the problems relating to rice production and makes workable suggestions for solutions.

Answer: Thanks for the strong support for publication.

Here are specific comments and suggestions, kindly address the following:

1. Mention the methodology used in the research work in the abstract and make the abstract self-explanatory, when read it in isolation.

Answer: Following this suggestion, we added text on the methodology (L 25-26 of revised MS).

2. Please include the previous work regarding the rice intensification studies in introduction.

Answer: Thanks. We included references of previous work on rice intensification (L 78-79 of revised MS).

3. Line 80-85, point out the reasons behind the low yield despite having an increase in cultivation area.

Answer: Thanks for this comment. We have already elaborated on the agronomic causes for the low yield (including poor water, fertilizer, and weed managements, varietal choice, and seed source, etc.) in the original MS (L 301-305 of revised MS). In response to this comment, and a previous one from Reviewer #2, we further elaborate in the revised MS regarding the socio-economic causes behind the low yield (L 313-314 of revised MS).

4. Mention the land conversion pattern in Africa with respect to rice crop over the study period in the introduction part of the manuscript.

Answer: Thanks for this comment. We already stated that the rice area has expanded by *ca.* 0.4 M ha per year over the past three decades in the original MS (L 48 of revised MS). Hence, no change was made in the revised MS in response to this comment.

5. Fig 2 is not visible?- Some issue- Kindly check.

Answer: Checked. Figure 2 is now visible in the revised MS.

6. Line number 112-113, Explain the process-based crop simulation models methodology you used in this study.

Answer: Following this comment, we added text on modeling (L 120 & L 122-124 of revised MS).

7. Line number 45 & 118, what is *ca.*?

Answer: Abbreviation for 'circa' (about in Latin), which is used widely in scientific articles.

8. Line number 390-391, Generally, for climatic studies minimum of 30 years are considered. Then why only 20 years has been considered?

Answer: Thanks for this comment. The goal of our study was to determine the average yield potential for each site. We agree that one needs to have a reasonable of years to account for the effect of year-to-year variation in weather on yield potential. Previous studies (Van Wart et al., 2013; Grassini et al., 2015) have analyzed how many years of weather data are needed to derive a reliable estimate of yield potential using crop simulations models and have found that that 10 years are sufficient for crops grown in favorable environments, as it is the case of irrigated and lowland rainfed rice in our study. In the case of unfavorable environment, as it is the case of upland rice in our study, the duration increases to 15 to 20 years. Hence using 20 years of weather data seems sufficient for a robust estimation of average yield potential and its stability. Based on this comment, we added text to the Methods section justifying the choice of number of years (L 453-460 of revised MS).

9. Line number 425, it will be helpful for readers if you could explain the methodology for calibration and validation of ORYZA model.

Answer: Done (L 137-140, L 508-514, L 519-529, L 546-549 & L 556-564 of revised MS, Supplementary Table 4 of revised SI).

10. Crop management – How the difference in crop management has been considered.

Answer: Thanks for this comment. Because yield potential depends on climate, including solar radiation, temperature, and water supply in the case of rainfed crops, it is important to simulate yield potential in the context of current crop calendars (as determined by crop establishment date & method, crop cycle length) and environment (irrigated, rainfed lowland, and rainfed upland). These data were compiled by personnel from AfricaRice and national research institutes, along with inputs from extension agents. Based on this comment, we added this explanation to the MS (L 102-104 & L 477-485 of revised MS). A detailed description on crop management is shown in the Supplementary Information (Text Section 5 of revised SI).

11. SSR- How it has been extrapolated to the regional level, which needs to be explained.

Answer: Thanks for this comment. Current rice demand and production were estimated for all countries in Africa, then summed up, and finally the SSR was calculated as the ratio between production and demand. This was already explained in the original MS (L 616-617 & L 650-653 of revised SI). For annual rice production by year 2050, we summed up the production from the 15 selected countries and we added that from the other rice producing countries in Africa (additional 43 countries and disputed territories), assuming for the latter that the average yield gap is the same as for the other 15 countries (L 660-662 of revised SI). In the case of demand by 2050, in addition to the 15 countries that were selected in our analysis, we estimated future rice demand for other 22 African countries included in USDA (per-capita demand baseline) and IMPACT (relative change in per-capita demand) databases. We noted that the remaining 21 countries and disputed territories accounted for 5% of annual rice demand estimated for the above-mentioned 37 countries (15 selected countries in our study plus other 22 countries included in both USDA and IMPACT databases). Therefore, we multiplied the estimated annual rice demand from the 37 countries by 1.05 to determine the total annual rice demand in Africa (L 653-658 of revised MS). We note that our selected countries account for 80% of rice production in Africa; thus, the uncertainty for not including all countries is relatively small. We added a sentence on this to the Methods section (L 659-660 of revised MS).

12. Suggest to include a conclusion section, with suggestions and strategies for future directions.

Answer: Nature Communications does not allow a Conclusion section. Hence, we added text on suggestions and strategies for future directions in the Discussion section of the revised article (L 346-357 of revised MS).

13. Uncertainties involved in the study and possible strategies to overcome them.

Answer: Following the suggestion, we added an uncertainty section with possible strategies to overcome the uncertainties (L 359-398 of revised MS).

14. Improving yields through sustainable and environmentally friendly practices, utilizing existing cropland efficiently, and carefully considering the impacts of cropland expansion. Ultimately, a combination of these strategies, alongside investments in agricultural research and infrastructure, is likely the most viable path to ensuring food security while minimizing negative environmental and social consequences- So these points needs to be focused in the revision.

Answer: Thanks. We added text to emphasize these points (L 280-281 & L 346-357 of revised SI).

Reviewer #2 (Remarks to the Author):

The authors have addressed most of my concerns successfully, and we have all learned something in the process. I believe the quality of the article has improved now.

Responses to Reviewers' Comments (and our responses **in red)**

We thank the editor and reviewers for their strong support for publication. We have addressed all their comments in the revised MS. Our point-by-point responses to the reviewers' comments are shown below in red.

Reviewer #2 (Remarks to the Author):

The authors have addressed most of my concerns successfully, and we have all learned something in the process. I believe the quality of the article has improved now.

Response: Thanks